# Pain in Patients with Post Paralytic Hemifacial Spasm: Before, during and after Botulinum Toxin Injections

**DOI:** 10.3390/toxins14010020

**Published:** 2021-12-27

**Authors:** Chloé Wehrlin, Diane Picard, Frederic Tankéré, Rémi Hervochon, Claire Foirest

**Affiliations:** ENT Department, Pitié Salpêtrière Hospital, Sorbonne Université, 75013 Paris, France; diane.picard@aphp.fr (D.P.); frederic.tankere@aphp.fr (F.T.); remi.hervochon@aphp.fr (R.H.); claire.foirest@aphp.fr (C.F.)

**Keywords:** facial palsy, pain, botulinum toxin

## Abstract

It is well-established that botulinum toxin (BT) injections improve quality of life in patients with postparalytic hemifacial spasm. Nevertheless, injection-related pain and contracture-related pain have not yet been studied. The primary objective of our study was to evaluate injection-related pain in patients with facial palsy sequelae, and to compare the standard technique (syringe) with the Juvapen device. The secondary objective was to evaluate the improvement of contracture-related pain one month after BT injection. Methods: We conducted an observational, prospective, monocentric study based on 60 patients with facial palsy sequelae who received BT injections in our university ENT (ear, nose throat) department. There were 30 patients in the Juvapen group (J) and 30 in the standard technique group (ST). All patients completed Numerical Rating Scale (NRS) questionnaires immediately after the injections and one month later. Results: The average NRS score was 1.33/10 with Juvapen and 2.24/10 with the standard technique (*p* = 0.0058; Z = 2.75). In patients with contracture-related pain, the average NRS score was 3.53 before BT injection, and 0.41 one month after BT injection (*p* = 0.0001). Conclusions: Juvapen is a less-painful injection technique than the standard one. BT reduces contracture-related pain one month after injection.

## 1. Introduction

Peripheral facial palsy incidence is about 0.2 per 1000 per year. One third of patients with peripheral facial palsy do not fully recover and have chronic sequelae, particularly postparalytic hemifacial spasm [1]. Additionally, facial palsy has a major psychosocial impact on patients [2]. Botulinum toxin (BT) injections reduce synkinesis, contractures and spasms on the affected side, and also allow symmetrization of the healthy side by reducing contralateral hyperactivity. In our clinical experience, we have observed that BT injections may reduce contracture-related pain, but to our knowledge, this has never been described in the literature. Moreover, BT injection-related pain in patients with facial palsy has never been studied.

Juvapen is a cordless motorized injection system produced by Juvaplus and intended to assist practitioners with injecting botulinum toxin. Both techniques, standard and Juvapen, use the same 30-gauge needle. Juvapen can be operated in two specific positions due to its ergonomic handle. Gentle pressure on its plastic button delivers a selected volume of botulinum toxin at a controlled speed. The speed and the depth are calculated in order to decrease the injection-related pain [3]. Nevertheless, pain reduction has never been studied in patients with facial palsy sequelae.

The primary objective of our study was to evaluate injection-related pain in patients with facial palsy sequelae, and to compare the standard technique (syringe) with the new Juvapen device. The secondary objective was to evaluate the improvement of contracture-related pain one month after BT injection.

## 2. Results

### 2.1. Characteristics of the J and the ST Groups

Sixty-two patients with post-paralytic hemifacial spasm received BT injections in our university ENT department between February and April 2021. All patients had either postparalytic hemifacial spasm (injection on the paralyzed side), flaccid palsy (injection on the healthy side), or both. We excluded two patients: one with facial diplegia and another who was treated for Frey syndrome. Thirty patients were injected with the standard technique (ST) and 30 patients were injected with the Juvapen device (J). The two groups were comparable (Table 1). There were no differences in the two treatment groups with regard to dose, dilution or volume injected. The doses injected varied between 2 and 4 IU (international unit) depending on the injection sites, but the two groups were comparable in terms of the number of injection sites (Table 1). The dilution was the same in both groups: 50 IU of botulinum toxin was diluted in 1.25 mL of injectable physiological serum.

### 2.2. Contracture-Related Pain before Botulinum Toxin Injections

Before injection, 16 patients reported pain related to muscle contractures (*n =* 9 in J group; *n* = 7 in ST group). The average NRS score was 3.53/10 before the injection in patients who reported contracture-related pain. No difference between the two groups was found: the average NRS score was 3.42/10 in the J group and 3.64/10 in the ST group.

### 2.3. Comparison of Injection-Related Pain in J and TS Groups

During the injection, the global average NRS score was 1.30/10 in the J group and 2.04/10 in the ST group (*p* < 0.0098; Z = 2.58) (Figure 1). On the affected side, NRS score was 1.33 in the J group and 2.24 in the ST group (*p* = 0.0058; Z = 2.75). On the healthy side there was no significant difference between the two injection techniques.

On the affected side the NRS score with J was significantly lower than with ST on peribucal and periocular territories (Table 2). There was no significant (NS) difference in the forehead, neck or cheek. When patients were interviewed after botulinum toxin injections, 17% of them wanted pain relief for future injections.

### 2.4. Contracture-Related Pain at One Month after BT Injections

Eighteen patients (30%) had contracture-related pain before botulinum toxin injections. Their average NRS-score was 3.53/10 before injection and 0.41/10 one month after injection (*p* = 0.0001; Z = 7.96). A total of 88.8% of patients with contracture-related pain before the injection reported being satisfied after BT administration.

## 3. Discussion

The Juvapen device caused less pain than the standard technique, especially on the affected side and on peribucal and periocular regions. Botulinum toxin injections improved contracture-related pain at one month after the injection.

Bertossi et al. have compared pain in two groups of patients in aesthetic medicine using the Juvapen device [3]. One group of 25 patients was injected using the standard technique and another group of 25 patients was injected using the Juvapen device. They showed a significant difference in terms of pain. Indeed, the VAS (visual analogic scale) score was 8/10 with ST and 3/10 with Juvapen device. We report similar results in terms of global pain, and we additionally found an enhanced benefit on periocular and peribucal regions. This new device allows greater precision that enables greater safety, comfort of use and better results. The electronic flow allows more accurate and reproducible injections with no product loss. Users estimate a 20% product savings.

Sarifakioglu et al. evaluated pain during cosmetic injections in 24 patients. Using the VAS scale, they compared direct injections to injections after a 5-min ice application. Pain was significantly lower on the side injected after ice application [4].

In our study, botulinum toxin injections reduced contracture-related pain. This is a well-known effect which has not been studied specifically on facial palsy sequelae. There is only one study that showed a long-term decrease in contracture-related pain after BT injections. Indeed, Dall’Angelo et al. studied a cohort of 69 patients with facial palsy treated for platysma’s synkinesis and reported relief of contractures and synkinesis for an average of 4 months. They also observed that the patients always reported improvement at each injection session, but that injections had to be repeated due to the temporary duration of the toxin’s effect [5]. BT injections are also used to reduce spasticity and pain in patients with neurological impairment of upper and lower limbs [6,7]. While functional improvement has been monitored, no study has reported potential decreases in pain after injections.

From our clinical experience, patients sometimes report a recurrence of contracture-related pain several months after BT injections. Accordingly, future research might evaluate the duration of contracture-related pain relief so that the time intervals between injections could be optimized, as this would directly improve patients’ quality of life. Further, a study on whether the benefits of BT injections on contracture-related pain remain stable or if the effect decreases after several years of repeated injections would be relevant. 

We led the first prospective study measuring BT-injection-related pain in patients with facial palsy. The main limitation is the pain evaluation scale used. Although the NRS is easy to use, it remains very subjective. We did not use other standardized scales in our protocol, such as quality of life scales or perception of sequelae, because we specifically focused on pain. However, many factors can influence quality of life—such as the etiology of the facial palsy, anxiety, chronic pain or being overweight—and can therefore also bias the patient’s feelings and pain tolerance during botulinum toxin injections. [8].

## 4. Conclusions

It is well known that BT injections allow symmetrization and reduce postparalytic hemifacial spasm. We demonstrated that BT is also an effective treatment for contracture pain in these patients. Pain during botulinum toxin injections has been studied in aesthetic medicine, but never in patients with facial palsy sequelae. Injection-related pain is reduced when using the Juvapen* device, especially on the affected side and on peribucal and periocular regions. We therefore suggest that Juvapen is a reliable device that can reduce pain during injections in patients with facial palsy sequelae, and we also suggest that BT injections can be used to reduce contracture pain in patients with postparalytic spasm.

## 5. Materials and Methods

We led an observational, prospective, monocentric study based on 60 patients with facial palsy sequelae who received BT injections in our university ENT department between February 2021 and April 2021. One experienced ENT practitioner injected all patients with 30 G needles. We compared two different injection techniques: the standard manual technique (0.5 mL syringe) vs. the Juvapen device. The first thirty patients were injected using the standard technique and the last thirty received Juvapen. The BT used in the study were Botox 100, Botox 50 (Allergan, Courbevoie, France), Xeomin 100 and Xeomin 50 (Merz, Courbevoie, France).

### 5.1. Population

We included patients older than 18, with synkinesis sequelae of facial palsy or post-paralytic hemifacial spasm. We excluded patients with Frey syndrome or hemifacial spasm due to vascular-nervous conflict, and patients with facial diplegia. All patients gave their informed and written consent according to IRB approval number 20210322165101 (APHP registry).

### 5.2. Data Collection

The following data were collected: demographics (age, gender, BMI (body mass index) and etiology of facial palsy); BT injection territories (forehead, eye, cheek, mouth and neck); and Numerical Rating Scale (NRS) pain score from 0 to 10 before, during, and after (one day and one month) injections. Patients were specifically queried about their contracture pain. Patients were also asked if they would desire analgesic drugs for their next injections. Patients were blinded to the injection technique and, in fact, were not aware there were different injection techniques being used in the department. During the collection of the NRS, double-blindness was not an option as the injector was aware of the injection technique.

### 5.3. Juvapen Device

The Juvapen is a cordless, motorized injection system produced by Juvaplus for use by practitioners to inject botulinum toxins. It can be used with a traditional 0,5 mL syringe and 30 to 35-gauge needles. Juvapen can be operated in two specific positions due to its ergonomic handle. Gentle pressure on its plastic button delivers a selected volume of BT at a controlled speed. The speed can be adjusted in order to decrease the injection-related pain and minimize toxin loss.

### 5.4. Statistics

Our main objective was to compare the pain during injection using the standard technique relative to the pain experienced using the Juvapen device. For this, we compared the pain’s global score by averaging the NRS of all injected territories. Therefore, we had two independent samples that did not follow a normal distribution. The data were nonparametric, and we used a Wilcoxon test to compare the two groups. We then compared the two groups based on the average pain in the affected areas after injection, and subsequently in healthy areas that underwent injection. On affected territories that had been injected, we also compared the pain by region (forehead, eye, cheek, mouth and neck). The secondary objective was to compare contracture pain before injection versus pain one month after injection. For this, we used a Wilcoxon test for paired magnitudes. For all tests, significance was considered if *p* < 0.001 and suggestive if *p* < 0.05 [9].

## Figures and Tables

**Figure 1 toxins-14-00020-f001:**
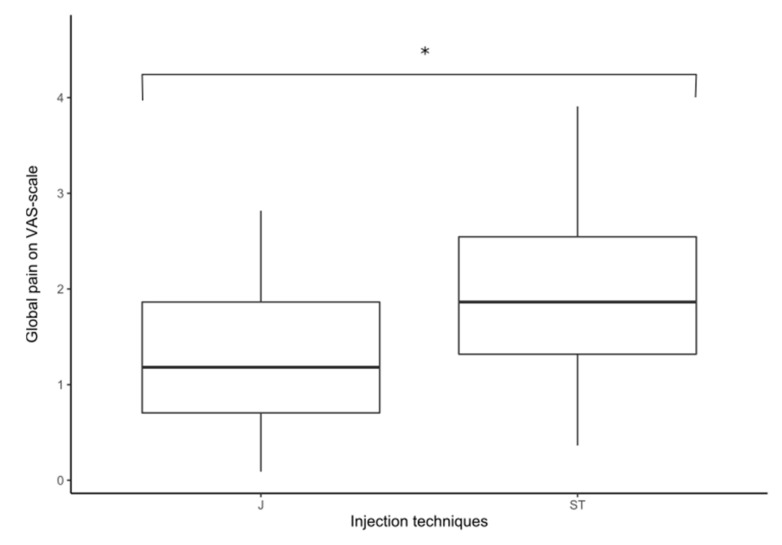
Global pain according to injection technique (* means that *p* < 0.05).

**Table 1 toxins-14-00020-t001:** Characteristics of patients who received botulinum toxin injections using the standard technique (ST) versus the Juvapen technique (J).

	J	ST
Age (years)	49.7	54.4
Women	53%	55%
Men	47%	45%
BMI	23.4 +/− 3.95	25.03 +/− 4.46
Infectious etiology	63%	66%
Traumatic etiology	37%	34%
House-Brackmann score	3.2 +/− 1.5	3.3 +/− 1.6
Botox	27%	10%
Xeomin	73%	90%
First-time injection	29%	27%
IU injected (average +/− standard deviation)	23.6 +/− 15.4	24 +/− 15.7
Number of injection locations (average +/− standard deviation)	8.43 +/− 4.3	8.3 +/− 4.8
Contracture-related pain	30%	24%

**Table 2 toxins-14-00020-t002:** Comparison between ST and J on each territory (NS means non-significant).

Territory	NRS J	NRS ST	*p*-Value
Forehead	2.64	2.32	NS
Eye	1.87	4.75	0.029
Cheek	3.16	2.1	NS
Mouth	1.82	3.61	0.0035
Neck	3.08	1.96	NS

## Data Availability

The data presented in this study are available upon request from the corresponding author. The data are not publicly available.

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
