# Peer review of "Pain in Patients with Post Paralytic Hemifacial Spasm: Before, during and after Botulinum Toxin Injections"

_toxins, 2021, doi:10.3390/toxins14010020_

Round 1
Reviewer 1 Report
This study is interesting. However, the authors should write more detail on Materials and Methods, and write more results.
- It has been already reported that Juvapen device gives less injection-related pain. The authors should write more regarding importance of this current study.
- The authors should write how they decided to use Juvapen device or standard syringe. It should be randomized.
- The authors should write the injection was in blind condition. The patients and evaluator should not know if the patients were which group.
- The authors should report if there was difference on the facial palsy between Juvapen device and standard syringe.
- The authors should write detail what the spasm related pain is. Also, the authors should report separately with Juvapen device and standard syringe on spasm related pain one month after injection.
Author Response
- It has been already reported that Juvapen device gives less injection-related pain. The authors should write more regarding importance of this current study. Thank you, a new chapter has been added in the introduction
- The authors should write how they decided to use Juvapen device or standard syringe. It should be randomized. Thank you. There was no randomization. The distribution of patients in both groups is now explained in the first chapter of materials and methods.
- The authors should write the injection was in blind condition. The patients and evaluator should not know if the patients were which group. Thank you. Blind conditions were impossible in this study : both patients and practitioner knew in which group the patient was. Indeed, the juvapen device is morphologically and functionally totally different from a standard syringe.
- The authors should report if there was difference on the facial palsy between Juvapen device and standard syringe. Thank you. Caracteristics of facial palsy are described in table 1 (demographic data, etiology, pain).
- The authors should write detail what the spasm related pain is. Thank you. Corrections have been performed. We wrote contracture-related pain instead of “spasm-related pain”
- Also, the authors should report separately with Juvapen device and standard syringe on spasm related pain one month after injection. Thank you. Considering that it is the same product and the same volume injected, for the same indication, we deliberately did not compare the pain at 1 month separately in group J and group ST. Also, the calculation would be biased given that the percentage of patients with contracture pain is initially different in both groups (cf table 1).

Reviewer 2 Report
The manuscript aims to demonstrate effect of Botulinum toxin injections to pain related to facial palsy.
My comments are as follows:
- The title does not reflect the study
- The study compares just two different methods for administration by standardized method or with Juvapen
- The studied groups had few patients with pain (30%/24%)
- The studied patients had not significant pain (score below 3 in both groups)
CONCLUSION: this study so not contribute to new knowledge about Botulinum toxin
Author Response
- The title does not reflect the study : Thank you. The new title is “Pain in Patients with post paralytic hemifacial spasm : before, during and after Botulinum Toxin Injections”
- The study compares just two different methods for administration by standardized method or with Juvapen : Juvapen is a new device which has never been tested in patients with facial palsy sequelae. Moreover, to our knowledge, Botulinum toxin effects on Contracture-related pain has never been studied.
- The studied groups had few patients with pain (30%/24%) : Thank you. We agree with the reviewer : most patients do not suffer from contracture-related pain.
- The studied patients had not significant pain (score below 3 in both groups) Thank you. We agree with the reviewer : The pain associated with post-paralytic contracture is moderate.
Reviewer 3 Report
The term "Spastic Sequelae Following Facial Palsy" is misleading. Please change throughout the manuscript. Correct is "(hemifacial) spasm" as this peripheral disorder has nothing to do with spasticity.
Please define the term "spastic pain". In hemifacial spasm pain is an extremely rare and atypical complaint of patients. What is the evidence for this oberservation?
Where there any differences in the two treatment groups with regard to dose, dilution or volume injected? Please provide data.
Results, row 40: Do you mean "facial palsy" or " hemifacial spasm"? Please define. Facial palsy does not need BoNT injections.
Results: It is not clear when the authors refer to pain before, during or after injection although they describe it in the Methods section. This is misleading.
Discussion rows 96/97: there are many studies on reduction of pain after BoNT injections in various neurological disorders.
Author Response
- The term "Spastic Sequelae Following Facial Palsy" is misleading. Please change throughout the manuscript. Correct is "(hemifacial) spasm" as this peripheral disorder has nothing to do with spasticity. Thank you very much. Word “spasticity” and "Spastic Sequelae Following Facial Palsy" have been totally removed. “Post paralytic hemifacial spasm” is know employed. The title has also been modified.
- Please define the term "spastic pain". In hemifacial spasm pain is an extremely rare and atypical complaint of patients. What is the evidence for this oberservation? Thank you. We are dealing with contracture related pain which affect 24 to 30% of our patients (table1). This has been modified
- Where there any differences in the two treatment groups with regard to dose, dilution or volume injected? Please provide data. Thank you. Additional data has been added to Table 1 and to Result chapter. “There were no difference in the two treatment groups with regard to dose, dilution or volume injected. The doses injected varied between 2 and 4 IU (international unit) depending on the injection sites, but the two groups were comparable in terms of the number of injection sites (table 1). The dilution was the same in both groups : 50 IU of botulinum toxin were diluted in 1.25ml of injectable physiological serum.” :
- Results, row 40: Do you mean "facial palsy" or " hemifacial spasm"? Please define. Facial palsy does not need BoNT injections. : Thank you. It is modified : “post paralytic hemifacial spasm”
- Results: It is not clear when the authors refer to pain before, during or after injection although they describe it in the Methods section. This is misleading. Thank you. Pain before injection is at the end of table 1. Pain during injection is chapter 2.2. Pain after injection is chapter 2.3
- Discussion rows 96/97: there are many studies on reduction of pain after BoNT injections in various neurological disorders. Thank you. Some references have been added.
Round 2
Reviewer 1 Report
Thank you for your response. The authors can not evaluate this device is good because the study plan has fault.
Author Response
Thank you very much. This chapter has been added.
For both techniques, we use the same 30 gauge-needle.Juvapen can be handled with two specific positions having an ergonomic handle. A gentle pressure on its plastic button delivers at controled speed a selected volume of botulinum toxin. The speed and the depth are calculated in order to decrease the injection-related pain.
Reviewer 2 Report
Please see the attachment
Author Response
1. The autors have compared two groups on this study :1) treated by standardized method with botulinum toxin and 2) trested by Juvapen. Why ? The information about differences and advantages of these 2 methods should been described in the Introduction in short manner.
Thank you very much. This chapter has been added.
For both techniques, we use the same 30 gauge-needle.Juvapen can be handled with two specific positions having an ergonomic handle. A gentle pressure on its plastic button delivers at controled speed a selected volume of botulinum toxin. The speed and the depth are calculated in order to decrease the injection-related pain.
2. The authors claim that Juvapen caused less pain. However, both methods did not cause much pain. Pleaseclarify
We agree with the reviewer. Thank you.
Indeed, both techniques are not very painful but the juvapen is even less painful than the standard one.Patients with post-paralytic sequelae have a difficult experience. That’s why we wanted to improve the injection sessions as much as possible. We are pleased to notice that some patients have an experience pain at 0/10 when injected with Juvapen device.
3. In the Discussion (line 121-160) has been discussed effect of the analgesic gels. This examples seem to be out of scope.
Thank you
. Sentence dealing with analgesic gels have been removed.
Reviewer 3 Report
no comments
Author Response
Thank you very much for your answer
Round 3
Reviewer 1 Report
I think that there are some problems in this study.
Author Response
Thank you for your answer
Reviewer 2 Report
The revised manuscript has been improved according to the previous comments. However, the conclusion should be adjusted because pain, investigated in this study, is not pronounced measured by NRS. Please rewrite the conclusion. I think that it is fair to state that if the hemifacial spasm coincides with or causes pain, the relief of pain may also may be expected.
Author Response
Thank you very much for your answer.
This is the new conclusion (modified in the manuscript) :
"
It is well known that BT injections allow symmetrization and reduce postparalytic hemifacial spasm. We demonstrated that BT is also an efficient treatment for contracture pain in these patients. Pain during botulinum toxin injections has been studied in aesthetic medicine, but never in patients with facial palsy sequelae. Injection related pain is reduced when using Juvapen* device, especially on the affected side and on peribucal and periocular regions. We therefore suggest that Juvapen is a reliable device that can reduce pain during injections in patients with facial palsy sequelae and we also suggest that BT injections can be used to reduce contracture pain in patients with postparalytic spasm.